# Current Status and Prospects of Immunotherapy for Gynecologic Melanoma

**DOI:** 10.3390/jpm11050403

**Published:** 2021-05-12

**Authors:** Mayuka Anko, Yusuke Kobayashi, Kouji Banno, Daisuke Aoki

**Affiliations:** Department of Obstetrics and Gynecology, Keio University School of Medicine, 35 Shinanomachi, Shinjuku-ku, Tokyo 160-8582, Japan; may.an0625@keio.jp (M.A.); kbanno@z7.keio.jp (K.B.); aoki@z7.keio.jp (D.A.)

**Keywords:** melanoma, mucosal melanoma, gynecologic melanoma, vulvovaginal melanoma, immunotherapy, immune checkpoint inhibitor, targeted therapy, KIT, imatinib

## Abstract

Gynecologic melanomas are rare and have a poor prognosis. Although immunotherapy (immune checkpoint inhibitors) and targeted therapy has greatly improved the systemic treatment of cutaneous melanoma (CM) in recent years, its efficacy in gynecologic melanomas remains uncertain because of the rarity of this malignancy and its scarce literature. This review aimed to evaluate the literature of gynecologic melanomas treated with immunotherapy and targeted therapy through a PubMed search. We identified one study focusing on the overall survival of gynecologic melanomas separately and five case series and nine case reports concentrating on gynecologic melanomas treated with an immune checkpoint inhibitor and/or targeted therapy. Furthermore, the *KIT* mutation has the highest rate among all mutations in mucosal melanoma types. The KIT inhibitors (Tyrosine kinase inhibitors: TKIs) imatinib and nilotinib could be the treatment options. Moreover, immune checkpoint inhibitors combined with KIT inhibitors may potentially treat cases of resistance to immune checkpoint inhibitors. However, because of the different conditions and a small number of cases, it is difficult to evaluate the efficacy of immunotherapy and targeted therapy for gynecologic melanoma rigorously at this time. Further prospective cohort or randomized trials of gynecologic melanoma alone are needed to assess the treatment with solid evidence.

## 1. Introduction

Gynecologic melanomas are uncommon. They stand for 18% of mucosal melanoma (MM; originating primarily from the head and neck, and anorectal and vulvovaginal regions), which accounts for 1% of all melanomas. The most common site is the vulva (70%), vagina, and the cervix of the uterus, in that order. Gynecologic melanomas also have a poor prognosis, with a 5-year overall survival (OS) of 37–50%, 13–32%, and approximately 10% for vulvar, vaginal, and cervical melanomas, respectively [1].

Meanwhile, the systemic therapy for cutaneous melanoma (CM) has improved recently by immunotherapy (immune checkpoint inhibitors) and targeted therapy; this therapeutic approach inhibits the mitogen-activated protein kinase (MAPK) pathway including BRAF, NRAS, MEK, and KIT. In almost fifty percent of patients with metastatic melanomas, valine is replaced with glutamine in codon 600 of the serine/threonine kinase BRAF (BRAF V600 mutation) [2,3,4]. However, BRAF inhibitors can cause the increase of melanocyte differentiation antigens, antigen-specific T cell recognition, CD8^+^ T cell infiltration, and T cell exhaustion markers (e.g., TIM3, PD1, and PDL1). When the MAPK pathway is reactivated, melanoma antigens are suppressed, and an immunosuppressive tumor microenvironment reemerges. If this pathway is inhibited subsequently by an MEK inhibitor, melanoma antigens are restored, and CD8^+^ T cell infiltration is promoted [5,6,7].

According to the National Comprehensive Cancer Network Guidelines version 2 (2021) [8], the following drugs are recommended for the systemic therapy of metastatic or unresectable CM. If the tumor has *BRAF* V600-activating mutation, BRAF inhibitor combined with MEK inhibitor is preferred (e.g., dabrafenib/trametinib, vemuragfenib/cobimetinib, and encorafenib/binimetinib; category 1, high-level evidence). If no mutation is observed, anti-PD-1 monotherapy or combination therapy including pembrolizumab/nivolumab or nivolumab plus ipilimumab (a monoclonal antibody for anti-cytotoxic T-lymphocyte–associated protein 4; CTLA-4) (category 1, high-level evidence) is preferred. Other recommended options (category 2B) are pembrolizumab/low-dose ipilimumab, a combination of targeted therapy and immunotherapy (e.g., vemurafenib/cobimetinib plus atezolizumab, dabrafenib/trametinib plus pembrolizumab). The maximum efficacy of these drugs in clinical trials with patients with CM is beyond the scope of this article.

In terms of biological characteristics, MM and CM seem to be different. MMs have a considerably lower rate (3–15%) of BRAF gene mutations than CM without chronic sun damage [9,10,11]. In addition, increased *BRAF* gene mutation in MM affects gene regions other than codon 600 or is a non-activating mutation, thereby not predicted to respond to targeted BRAF inhibitor [10]. On the contrary, gene copy number and structural variations, such as in KIT, are considerably more frequent in MM than in CM [12].

In this regard, KIT inhibitors (Tyrosine kinase inhibitors: TKIs) relevant to the treatment of MM include those listed in Table 1 [13,14,15,16,17,18,19,20,21]. These drugs are mainly administered in CML, but some trials demonstrated the efficacy of *KIT* mutated melanoma, including MMs.

With regard to immunotherapy, randomized clinical trials of ipilimumab, pembrolizumab, and nivolumab for MM are currently unavailable, but these medicines are effective according to some studies. Table 2 summarizes the results of these trials [12,22,23]. Of note, in a pooled analysis of five clinical trials, the efficacy seemed to be greater in nivolumab plus ipilimumab than in either of the agent alone, and the activity was lower in MM than in CM [12].

However, the efficacy of immunotherapy and targeted therapy in gynecologic melanomas is uncertain because this malignancy is rare, thereby obtaining scarce literature. Hence, this review aimed to evaluate the literature of gynecologic melanomas treated with immunotherapy and targeted therapy, and consider the possibility of such treatment in the future.

## 2. Materials and Methods

We searched the literature in PubMed using the following search strings:

“melanoma” AND “genital” AND “immunotherapy”; “MM” AND “genital” AND “ipilimumab”; “melanoma” AND “genital” AND “nivolumab”; “melanoma” AND “genital” AND “pembrolizumab”; “melanoma” AND “vulvovaginal” AND “ipilimumab”; “melanoma” AND “vulvovaginal” AND “nivolumab”; “melanoma” AND “vulvovaginal” AND “pembrolizumab”; “melanoma” AND “vulvovaginal” AND “immunotherapy”; “melanoma” AND “gynecologic” AND “immunotherapy”; “melanoma” AND “gynecological” AND “immunotherapy”; “melanoma” AND “vagina” AND “immunotherapy”; “melanoma” AND “uterine cervix” AND “immunotherapy”; “melanoma” AND “vulva” AND “immunotherapy”; “MM” AND “immune checkpoint inhibitor”; “melanoma” AND “genital” AND “targeted therapy”; “melanoma” AND “vulvovaginal” AND “targeted therapy”; “melanoma” AND “vulva” AND “targeted therapy”; “melanoma” AND “vagina” AND “targeted therapy”; “melanoma” AND “uterine cervix” AND “targeted therapy”; “melanoma” AND “gynecologic” AND “targeted therapy”; “melanoma” AND “gynecological” AND “targeted therapy”; “Mucosal melanoma” AND “tyrosine kinase inhibitor.”

### 2.1. Inclusion and Exclusion Criteria

We included articles that focused on human studies exclusively, used the English language, and were published between 10 years ago and March 2021. We excluded articles concerning CMs, MMs of non-gynecologic organ origins, and gynecologic tumors other than melanoma. In the case of vulvovaginal melanomas, one question about the distinction between mucosal and cutaneous melanoma might have occurred. The location of the tumor from the vulva or vagina, specifically within X cm, was not mentioned in any literature. However, as it is assumed that gynecologic melanoma is included in mucosal melanoma in general, we treated all of them as mucosal melanoma.

### 2.2. Literature

In total, we found 252 articles in PubMed. Among them, six studies and nine case reports were included.

## 3. Results

We identified five case series and nine case reports that focused exclusively on gynecologic melanomas treated with immune checkpoint inhibitors and/or targeted therapy (Table 3). In addition, one study assessed the OS of gynecologic melanomas separately, along with MM.

A population-based study in the Netherlands evaluated the benefits of immune checkpoint inhibitors to MM in comparison with those to CM in patients with advanced melanoma between 2013 and 2017 [24]. CM and MM were staged according to the American Joint Committee on Cancer (AJCC) staging classification [25]. 2960 patients with CM and 120 patients with MM who registered in the Dutch Melanoma Treatment Registry were retrospectively analyzed. Among them, 29 (24%) had MM in the vulvovaginal region. Initial immunotherapy and targeted therapy were given to 77% and 2% of patients with MM in comparison with 49% and 33% of patients with CM, respectively. The median OS was 8.9 months (95% CI: 7.3–12.7) and 14.5 months (95% CI: 13.7–15.4) for patients with MM and with CM respectively. For vulvovaginal melanoma, the median OS was 8.6 months (95% CI: 6.8–21). Interestingly, the median OS of patients with MM diagnosed in 2013–2014 and 2015–2017 was almost the same, 8.7 months (95% CI: 6.9–16.7) and 8.9 months (95% CI: 6.8–13.5), respectively, while the median OS of patients with CM increased from 11.3 months (95% CI: 10.2–12.4) to 16.9 months (95% CI: 15.4–18.2), respectively. Furthermore, the frequency of oncogenic mutations was lower in MM than in CM: BRAF mutations were detected in 1649 (55.9%) patients with CM and in 7 of 122 patients with MM (5.8%; 5 V600E, 1 V600R, 1 V600K, and 1 “other”). In addition, 625 (21.1%) and 39 (1.3%) patients with CM and 17 (14.2%) and 15 (11.7%) patients with MM showed *NRAS* and *KIT* mutations, respectively. However, *KIT* was most frequently mutated in four (13.8%) patients with vulvovaginal melanoma.

We identified five retrospective case series. Table 3 is based on Table 3 in the study by Wohlmuth et al. [26]. Adding other references’ patients with gynecologic melanomas treated with immune checkpoint inhibitors. Treatment response was assessed retrospectively using the Response Criteria for Use in Trials Testing Immunotherapies (iRECIST) [27].

AJCC staging classification (eighth edition) was applied for vulvar melanomas, and vaginal or cervical melanomas were classified as local, regional, or distant [25]. In one article, Ballantyne’s staging system, which had been used since the 1970s for staging MM, was used. Luna-Ortiz et al. [28] reported that Ballantyne’s localized disease is approximately equal to moderately and very advanced disease in the AJCC staging classification.

In a single-center study in Toronto over 15 years, 13 patients with advanced MMs were treated with immune checkpoint inhibitors [26]. Their best overall ORR was 30.8% (95% CI: 5.7–55.9%), and the clinical benefit rate (CBR) was 61.5% (95% CI: 35.1–88.0%). The median PFS and the median OS were 4.0 months (95% CI: 2.3–5.7) and 17.0 months (95% CI: 12.7–21.3), respectively.

In another single-center study conducted in Milan between January 2011 and December 2016, seven patients who received immunotherapy for metastatic gynecologic melanomas were included [29]. The PFS and OS of patients treated with anti-PD-1 agents were better than those of patients treated with anti-CTLA4 agents (*p* = 0.01, log-rank test and *p* = 0.15, log-rank test: The *p*-value of OS was not statistically significant). The response rate to immunotherapy was 28.5%.

In a skin cancer department in France between 2013 and 2018, a study including 15 patients with unresectable or metastatic vulvovaginal melanoma was conducted [30]. Six patients were treated with ipilimumab. One melanoma carried a *NRAS* mutation (p.Q61R), and another melanoma carried a *KIT* mutation (exon 11). According to the OS achieved, 4 patients (66%) had a confirmed progressive disease (iCPD), 1 had an iSD (maintained for 11 months), and 1 had a good response in which the tumor burden was reduced by 89% and a long survival was achieved (31 months). The 1-year survival rate was 33%. Meanwhile, eight patients received nivolumab. One melanoma carried a *BRAF* V600E mutation (p.Q61R), and two melanomas carried a *NRAS* mutation (p.G12D). The best OS response was the iPR in four patients; the four other patients had an iCPD. The 1-year survival rate was 86%. One patient was treated with ipilimumab plus nivolumab as part of the clinical protocol CA209511. After 5 months from treatment initiation, an iPR (30% decrease in the target region) was obtained.

In another single-center study conducted between January 2006 and September 2013, 64 non-CM cases were collected, including eight vulvovaginal melanomas [31]. Tumor mutation was analyzed in 7 of 8 patients. Four samples were positive for *KIT* (4/7) mutations, including 2 exons of 13 K642, 1 exon of 17 N822K, and 1 exon of 11 L576P mutations. However, no *BRAF* (0/5) or *NRAS* (0/2) mutations were detected. Ipilimumab, pembrolizumab, interferon pegintron (100 μg/week, subcutaneous injection), imatinib, and nilotinib were administered to 3, 1, 1, 3, and 1 patients, respectively. Imatinib and ipilimumab showed the best response, resulting in iPR in 2/3 (66%) of cases with imatinib and 1/3 (33%) with ipilimumab. However, only one patient who received pembrolizumab had an iSD after 3 months. Adjuvant treatment with pegintron at 100 µg/week subcutaneously for more than 5 years lead to a successful outcome in one patient of nodal relapse following lymph node dissection and remained disease-free.

**Table 3 jpm-11-00403-t003:** Case series and reports of gynecologic melanoma treated with immune checkpoint inhibitors in details, based on Table 3 in Wohlmuth et al.

Author (Year)	Pt. No	Origin	^1^ Stage at Treatment Initiation	Primary Systemic Therapy	Treatment	iBOR	^2^ PFS	irAEs	^3^ OS	Status
Wohlmuth (2020) [26]	1	Vulva	III C, unresectable	None	Pembrolizumab	iCPD	2	None	18	Alive with disease
	2	Vulva	IV (lung)	None	Ipilimumab + nivolumab	iSD	18	Uveitis G1,peripheral sensory neuropathy G3	18	Alive with disesase
	3	Vulva	IV (liver)	None	Ipilimumab + nivolumab	iCPD	1	None	1	Died of disease
	4	Vulva	IV (liver)	None	Nivolumab	iPR	15	Hepatitis G1	15	Alive with disease
	5	Vagina	Distant (brain)	Nivolumab, adjuvant	Pembrolizumab	iSD	4	None	16	Died of disease
	6	Vulva	IV (lung)	None	Ipilimumab	iCR	56	None	56	Alive with NED
	7	Vulva	IV (lung, liver)	Interferon, adjuvant	1. Ipilimumab 2. Pembrolizumab	iCPD iSD	3 4	Maculopapular exanthema G1, Hepatitis G1 None	17	Died of disease
	8	Vulva	IV (liver)	None	1. Ipilimumab 2. Pembrolizumab	iCPD iPR	3 9	Maculopapular exanthema G1 None	50	Alive with disease
	9	Vulva	IV (lung, brain)	Carboplatin/paclitaxel	Ipilimumab	iCPD	3	None	6	Died of disease
	10	Vulva	IV (lung)	Dacarbazine	1. Ipilimumab 2. Pembrolizumab	iCPD iCR	3 77	None Hyperthyroidism G2, DM G3,Erythema nodosum G1	87	Alive with NED
	11	Vulva	IV (lung, bone)	None	Ipilimumab	iCPD	1	None	1	Died of disease
	12	Vulva	IV (lung, abdomen)	Carboplatin/paclitaxel	1. Ipilimumab 2. Pembrolizumab	iCPD iCPD	3 3	None None	16	Died of disease
	13	Vulva	IV (lung, abdomen, soft tissue)	Carboplatin/paclitaxel	Ipilimumab	iSD	2	None	13	Died of disease
Indini (2019) [29]	1	Vulva	IV (lung)	CVD	Ipilimumab	iCPD	4	None	7	Died of disease
	2	Vulva	IV (lung, bone)	None	Pembrolizumab	iPR	10	Arthralgia G2, hypothyroidism G2	10	Alive with disease
	3	Vagina	Distant (liver)	None	Pembrolizumab	iCPD	2	None	4	Alive with disease
	4	Vagina	Distant (n.s.)	None	Nivolumab	iSD	4	Cutaneous rash G1	4	Alive with disease
	5	Vagina	Distant (liver, pancreas, soft tissues, bone)	None	Ipilimumab	iCPD	3	None	7	Died of disease
	6	Vagina	Distant (lung)	Dacarbazine	Ipilimumab	iCPD	3	None	18	Died of disease
	7	Cervix	Distant (lung, liver)	None	Ipilimumab	iCPD	2	None	2	Died of disease
^4^ Quéreux (2017) [30]	1	Vulva or Vagina	5: Distant (mucosa and/or lymph nodes) 1: Distant (liver)	2: None 3: Chemotherapy 1: Nivolumab	Ipilimumab	4: iCPD		1: Asthenia G1 1: Colitis G1 1: Rheumatoid arthritis G1 1: Colitis G3	n.s.	The survival rate at 1 year: 33%
	2									
	3									
	4									
	5					1: iSD	11		11	
	6					1: iPR	31		31	
	7	Vulva or Vagina	6: Distant (mucosa and/or lymph nodes) 2: None	4: None 1: Dacarbazine 1: BRAF and MEK inhibitors 2: Ipilimumab	Nivolumab	4: iPR 4: iCPD	n.s.	3: Asthenia G1 2: Maculopapular rush G1 2: Rheumatoid arthritis G2 1: Colitis G2	n.s	The survival rate at 1 year: 86%
	8									
	9									
	10									
	11									
	12									
	13									
	14									
	15	Vagina	Distant (mucosa, lymph nodes, lung)	None	Ipilimumab + Nivolumab	iPR	5	Asthenia G1, Hypothyroidism G2	12	Alive with disease
^4^ Del Prete (2016) [31]	1	Vulva or Vagina	7: n.s. (surgery for total excision) 1: n.s. (unresectable)	1: pegylated interferon			n.s.	n.s.	n.s.	
	2				3: Ipilimumab	1: iPR, 2: iPD				n.s.
	3				1: Pembrolizumab	iSD				Alive with disease
	4				1: Interferon pegintron	iSD				Alive with NED
	5				3: Imatinib	2: iPR, 1: iSD				n.s.
	6				1: Nilotinib	iCPD				n.s.
	7				5: Chemotherapy	1: iSD, 4: iCPD				n.s.
	8									
Schiavone (2016) [32]	1	Vagina	localized, Ballantyne I	None	Ipilimumab	iSD	38	Maculopapular rush G3, Diarrhea G1	38	Alive with NED
	2	Vagina	localized, Ballantyne I	None	Ipilimumab	iCPD	2	None	16	Died of disease
	3	Vagina	localized, Ballantyne I	None	Ipilimumab	iCR	20	None	20	Alive with NED
	4	Cervix	localized, Ballantyne I	None	Ipilimumab Pembrolizumab	iCPD iSD	9 n.s.	Diarrhea G3 n.s.	19 n.s.	Alive with disease
Anko (2020) [33]	1	Vagina	Distant (liver, lung, bone)	None	Nivolumab	iPR/iCR	17	Thyroiditis, n.s	17	Alive
	2	Cervix	II C (None)	None	(recurrence)Nivolumab	iCR	33	None	33	Alive with disease
Cocorocchio (2020) [34]	1	Vulva	IV (lung, lymph nodes) IV (brain, adrenal gland, lung, subcutaneous) IV (breast, lung, stomach)	None	Ipilimumab + Nivolumab Nivolumab + RT(brain) Avapritinib	iSD iCPD iPR	10 4 11	Hyperglycemia G4 Hemiparesis, n.s. Vasculitis G2, uveitis G2	40	Died of disease
Komatsu-Fujii (2019) [35]	1	Vagina	Distant (lung)	None	Nivolumab Pembrolizumab Ipilimumab	iCPD iCPD iCPD	n.s n.s. n.s.	n.s. n.s. n.s.	n.s.	Alive with disease
Yamashita (2019) [36]	1	Vulva	1,2. IV (liver, lymph nodes) 3,4. IV (liver, lymph nodes, lung, gall bladder, renal duct)	None	Nivolumab Ipilimumab + RT (liver) Dacarbazine Pembrolizumab + imatinib	iCPD iCPD iCPD iPR	n.s. n.s. n.s.	Maculopapular rush G2	n.s.	Alive with disease
Norwood (2019) [37]	1	Vagina	Regional	None	(recurrence) Ipilimumab + Nivolumab	iSD	n.s.	Maculopapular rush G3, Colitis G2, Hyponatremia G2, Headache G2	n.s.	Alive with disease
Kim (2018) [38]	1	Cervix	Distant (bone, spine, lung, lymph nodes)	None	(adjuvant therapy) Pembrolizumab	iCPD	0	Maculopapular rush, n.s.	9	Died of disease
Daix (2018) [39]	1	Vagina	Regional, unresectable	None	Nivolumab	iCR	8	Pruritus G1	8	Alive with NED
Nai (2018) [40]	1	Cervix	Distant (liver, kidney)	None	Ipilimumab + Nivolumab	iCPD	0	None	12	Died of disease
Inoue (2018) [41]	1	Vagina	Distant (brain)	None	Nivolumab	iCPD	2	Hepatitis G3	n.s.	Alive with disease

^1^ Stage: AJCC staging classification for vulvar melanoma, and local/regional/distantfor vaginal or cervical melanomas; ^2^ PFS: from treatment initiation to date of progression or death; ^3^ OS: from treatment initiation to date of last follow-up or death; ^4^ No details of patients order in the treatment (ipilimumab, nivolumab, etc.). The leftmost number before colon indicates the number of patients in each group.Abbreviations: CVD indicates cisplatin-vinblastin-dacarbazine; Cervix, melanoma of uterine cervix; DM, diabetes mellitis; G, grade; iCPD, comfirmed progressive disease; irAEs, immune-related adverse events; NED, no evidence of disease; n.s., not specified; RT: radiotherapy.

Moreover, a study retrospectively evaluated four patients treated with combined immunotherapy and radiation for gynecologic melanomas between 2012 and 2015 [32]. *BRAF*, *NRAS*, or *KIT* mutation did not occur in 3/4 patients, while genetic testing was not performed in the remaining one patient did. All four patients received at least three ipilimumab doses with concurrent external beam radiation. The OS was 16 months in one patient with recurrent vaginal melanoma, 20–38 months in patients with non-recurrent vaginal melanoma, and 19 months in one patient with recurrent cervical melanoma. Table 3 summarizes the results of case reports [33,34,35,36,37,38,39,40,41].

## 4. Discussion

This study evaluated one retrospective population-based study, five retrospective case series, and nine case reports with gynecologic melanomas treated with immunotherapy and/or targeted therapy. Because of the different conditions and a small number of cases, it would be inappropriate to statistically analyze them together. And it is difficult to evaluate the efficacy of immunotherapy and targeted therapy for gynecologic melanoma rigorously at this time. So, the aim of this article is to assess the findings currently available and consider the possibility of treatment in the future.

In one study [24], patients with vulvovaginal melanomas had an OS similar to that of the patient with average MM and a shorter OS than the patient with the average CM. Between 2013 and 2017, the OS of patients with MM did not improve despite administering novel therapies. In addition, *KIT* was the more frequently mutated gene in MM of the vulvovaginal region rather than in CM.

Of the 57 patients included in the case series and case reports listed in Table 3, six achieved iCR, while thirteen achieved iPR. One, three, and two patients who achieved iCR received pembrolizumab, nivolumab, and ipilimumab, respectively. For iPR, two, five, one, one, one, two and one patients received pembrolizumab, nivolumab, ipilimumab plus nivolumab, ipilimumab, pembrolizumab plus imatinib, imatinib, and avapritinib, respectively. According to Indini et al., PFS was statistically better in patients treated with anti-PD-1 than in patients treated with anti-CTLA4 [29]. Overall, anti-PD-1 antibodies seemed to be more related to better prognosis than anti-CTLA4 antibodies. The *KIT* mutation was also frequently detected in gynecologic melanomas.

Zeijl et al. reported that MM, including gynecologic melanoma, had a worse prognosis than CM [24], correlated with the results of D’Angelo et al. [12]. The poor prognosis of MM is possibly attributed to the low tumor mutation burden (TMB) (causing a low response to immune checkpoint inhibitors), the absence of targetable oncogenic drivers, the rich lymphatic and vascular supply, and the alleged biological aggressiveness [42,43,44]. In comprehensive genomic profiling by Johnson et al., 2% of over 2000 melanoma samples were MM. MM had a markedly lower TMB than CM [11]. Consistent with their finding, a whole-genome or exome sequencing of 10 MM samples showed that MM had a 5–10-fold lower mutation rate than CM [45]. In one retrospective study investigating the genomic profile of acral melanoma, MM, and vulvovaginal melanoma (sun-protected melanomas), vulvovaginal melanoma had a lower TMB than MM or CM. Vulvovaginal melanoma obtained an overall TMB average of 1.65 ± 1.22 nonsynonymous mutations/Mb and a median of 1.80 (0.17–3.8), whereas MM obtained 6.11 ± 13.18 and 2.05 (0–64.33), respectively. These TMBs are considerably smaller than 16.8 mutations/Mb reported by The Cancer Genome Atlas Program for CM in general [46].

As shown in the literature being evaluated, MM has a higher *KIT* mutation rate than CM (39% vs. <3%), but the increased rate is mainly due to the increased frequency of vulvovaginal and anorectal melanomas, not head and neck melanoma [47]. In one study, the copy numbers and protein expression of selected genes in 2304 malignant melanoma samples were assessed through in situ hybridization and immunohistochemistry. 51 vulvovaginal melanomas (14 vaginal and 37 vulvar melanomas) were compared with 2253 nongynecologic melanomas (2127 CM, 105 MM, and 21 acral melanomas). The *KIT* mutation was most common in vulvovaginal melanomas (22%) in contrast to 3% in CM (*p* < 0.001). Meanwhile, *NRAS* mutations were rare in vulvovaginal melanomas in contrast to CM (25.9%; *p* = 0.009) [48]. In another study evaluating 65 vulvovaginal cases, the *KIT* mutations were found in 18% of vulvar melanomas, but none in vaginal tumors. In immunohistochemistry, moderate or strong KIT protein expression was present in 30 cases, including all tumors with *KIT* mutations and 6 of the 7 tumors with *KIT* amplification [49]. In another retrospective study, six (22.2%) of 30 patients with gynecologic melanomas (vulva, vagina, and cervix) had *KIT* mutations, particularly in exon 11 [50]. The *KIT* mutation may also serve as a prognostic factor. Moreover, 95 patients with vulvar melanoma from 10 clinical institutions underwent molecular analysis by either targeted next-generation or direct sequencing. Detected mutations were in *KIT* (44%), *BRAF* (25%), *NF1* (22%), *TP53* (17%), *NRAS* (9%), and telomerase reverse transcriptase promotor (9%). In a univariate analysis, *KIT* mutations were notably related to a better PFS (hazard ratio: 0.29, *p* = 0.0013) [51]. Compared with vulvovaginal melanomas with *KIT* mutations, vulvovaginal melanomas with wild-type KIT mutations tended to express molecular markers suggestive of platinum resistance (ERCC1), alkylating sensitivity (MGMT), and anthracycline sensitivity (TOP2A) [48].

These results suggest that patients with low TMB and frequent *KIT* mutations in gynecologic melanomas should be administered with KIT inhibitors such as imatinib, nilotinib, and dasatinib. The details of these drugs are shown in Table 1. Especially in one trial with nilotinib, two patients who previously had either a PR or CR to imatinib achieved durable PRs to nilotinib 12.4 and 20 months individually; thus, nilotinib can overcome acquired resistance to imatinib [19]. Of note, the efficacy of a KIT inhibitor may vary in the subtype of *KIT* mutations. In one report, six responses to KIT inhibition were detected in tumors with L576P or K642E mutation. In the same report, some patients had V654A and D820Y mutations, which are known to cause resistance to imatinib in GIST [52]; hence, using imatinib, these patients experienced disease progression. Thus, the overall response rates were low in GIST with *KIT* mutations reported in this previous study.

With regard to gynecologic melanomas and KIT inhibitors, we found six patients treated with TKI in one case series and two case reports, as presented in Table 3. Imatinib achieved two iPRs and one iSD in one case series. Two of the patients carried K642E mutation and one carried L576P mutation, which all well responded to KIT inhibitors [31]. In one case report, a 34-year-old patient with vulvar melanoma who was previously treated with nivolumab, ipilimumab, and dacarbazine was administered with pembrolizumab (2 mg/kg). And after 2 weeks, oral imatinib (400 mg) daily was also started [36]. Within 4 weeks, various symptoms, such as loss of appetite and upper abdominal pain, improved, and the level of serous LDH was decreased from 1047 U/L to 153 U/L. Computed tomography revealed that all metastases (lung, gall bladder, and renal duct) significantly mitigated, decreasing to half the size, and almost all the liver metastases necrotized. One patient harbored a *KIT* mutation, that is, Del579 in exon 11, which is less frequent (2%). The efficacy may be attributed to the single effect of imatinib or sequential effect after PD-1 blockade. Compared with imatinib alone, the combination treatment of PD-1/PD-L1 blockade and imatinib leads to increased intertumoral CD8^+^ T cell proliferation and inflammatory cytokine production; hence, immune checkpoint therapy could increase the antitumor effect of imatinib by enhancing the function of T cell effectors [53]. In another case report, avapritinib was administered to a 47-year-old patient with vulvar melanoma who was previously treated with ipilimumab plus nivolumab and nivolumab plus cyber-knife radiotherapy to CNS metastasis [34]. Avapritinib (BLU-285) is a selective oral kinase inhibitor that treats imatinib-resistant GISTs by targeting KIT/PDGFRα activation loop mutants (exon17/18). The United States Food and Drug Administration approved avapritinib for treating unresectable or metastasis PDGFRα exon 18 mutant GISTs. After 16 weeks of avapritinib therapy, the patient achieved PR in the primary and metastatic sites (lymph nodal, right adrenal gland, lung, and subcutaneous metastasis) and SD in CNS metastasis. Although PR was maintained for 11 months, the patient was diagnosed with PD and died after 4 months. The *KIT* mutation was p.N822K in exon 17.

A phase I study demonstrated another possibility of the combination of an anti-PD-1 antibody (toripalimab) and an anti-vascular endothelial growth factor antibody (axinitib) [54]. A total of 33 Asian patients, including seven gynecologic melanomas (21.2%), were enrolled. Among them, 29 had chemotherapy-naïve MM in which 14 (48.3%; 95% CI: 29.4–67.5%) achieved an objective response, and the median PFS was 7.5 months (95% CI: 3.7–not reached). However, these data need to be validated in a large cohort.

### Limitation and Proposal

As mentioned a little before, we found no studies that solely compare MM and CM in terms of the evaluation index, such as PFS, OS, and 1-year survival. Currently, assessing the optimal treatment or its efficacy to gynecologic melanomas with solid evidence is difficult. The possible treatments are the extrapolations of MM and CM treatments. If conducting a prospective cohort or randomized trials is difficult because of the small number of cases, a system that can register individual cases (including information on primary sites, genetic variants, treatment options, etc.) should be established. Gynecologic melanomas are likely to be included as part of MM, and the treatment is evaluated together with the other MM types. In particular, genitourinary MM is most common in non-Hispanic white people [55]; thus, accumulating a database of cases on a regional o rational basis would be meaningful. Another limitation is the diversity of terms referring to gynecologic melanomas; these terms include “vulvovaginal melanoma,” “melanoma of lower genital tract,” and “gynecological melanoma.” This diversity makes access to necessary information more difficult. If possible, common technical terms for gynecologic melanomas should be standardized.

## 5. Conclusions

Gynecologic melanomas are rare and have a poor prognosis. Because of the different conditions and the small number of cases, it is difficult to evaluate the efficacy of immunotherapy and targeted therapy for gynecologic melanoma rigorously at this time. And its possible treatments are the extrapolations of MM and CM treatments. Regarding genetic mutation, *KIT* is the most frequently mutated gene. A KIT inhibitor such as imatinib or nilotinib could be the treatment of choice. Moreover, a combination of an immune checkpoint inhibitor and a KIT inhibitor may potentially treat cases of resistance to immune checkpoint inhibitors. Further prospective cohort or randomized trials of gynecologic melanoma alone are needed to assess the treatment with solid evidence.

## Figures and Tables

**Table 1 jpm-11-00403-t001:** Tyrosine kinase inhibitors (TKIs) relevant to the treatment of MM.

Name	TKI Generation	Target	Indication in Clinical	Trials (Author, Year)	Pt. No (MM No:%)	Results [95% CI]	Note
Imatinib	1st	KIT, BCR-ABL, ^1^ PDGFRA	^2^ CML, KIT-positive ^3^ GISTs, Philadelphia chromosome-positive ^4^ ALL etc.	Guo (2011) [15]	43 (11:25.6%)	PR (best response): 10 (23.3%)	exons 11 and 13 mutations predict the response to imatinib
				Hodi (2013) [16]	24 (17:70.8%)	^5^ ORR:29%	
Nilotinib	2nd overcome resistance of BCR-ABL mutants to imatinib	KIT, BCR-ABL inhibitory activity to *KIT* mutations (exons 9, 11,13)	CML	Carvajal (2015) [19]	19 (12: 63%)	(Premedicated Pt with imatinib) ^6^ TTP (months): 3.4 OS (months): 14.2	Nilotinib may overcome acquired resistance to Imatinib
				Lee (2015) [20]	42 (12: 28.6%)	ORR: 16.7% [5.4–28.0]	
Dasatinib	2nd Multi-kinase TKI	KIT, BCR-ABL, SRC family kinases	CML, Philadelphia chromosome-positive ALL	Kalinsky (2017) [21]	73 (38: 52%)	Pt with PR (3: 5.9%) did not have *KIT* mutations	superiority to imatinib was not shown

^1^ PDGFRA: platelet-derived growth factor receptor alpha; ^2^ CML: chronic myelogenous leukemia; ^3^ GIST: gastrointestinal stromal tumors; ^4^ ALL: acute lymphoblastic leukemia; ^5^ ORR: objective response rate; ^6^ TTP: time to progression.

**Table 2 jpm-11-00403-t002:** Studies demonstrating the efficacy of immune checkpoint inhibitors in advanced melanomas including MM.

Author (Year)	Pt. No (MM No:%)	Treatment	Results		
Hamid O (2018) [22] post-hoc analysis of KEYNOTE 001, 002, 006	1567 (84: 5%)	Pembrolizumab	ORR:19% [95% CI: 11–29%] median duration of response: 27.6 months		
Nathan P (2019) [23] single-arm phase II study (CheckMate 172)	1008 (63: 6.3%)	Nivolumab	median OS (months): 11.5 (MM), 25.3 (non-acral CM) 18-month OS rates: 31.5% (MM), 57.5% (non-acral CM)		
D’Angelo (2017) [12] a pooled analysis of five clinical trials	889 (86: 10%)	Ipilimumab, Nivolumab, Nivolumab plus Ipilimumab	(1) nivolumab	MM [95% CI]	CM [95% CI]
			median ^1^ PFS (months)	3.0 [2.2–5.4]	6.2 [5.1–7.5]
			ORR (%)	23.3 [14.8–33.6]	40.9 [37.1–44.7]
			(2) nivolumab plus ipilimumab		
			median PFS (months)	5.9 [2.8–not reached]	11.7 [8.9–16.7]
			ORR (%)	37.1 [21.5–55.1]	60.4 [54.9–65.8]

^1^ PFS: Progression Free Survival.

## Data Availability

No new data were created or analyzed in this study. Data sharing is not applicable to this article.

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
