# Peer review of "Current Status and Prospects of Immunotherapy for Gynecologic Melanoma"

_jpm, 2021, doi:10.3390/jpm11050403_

Round 1
Reviewer 1 Report
we did appreciate your work, in relation to a topic which is still under study, in an era of rapidly evolving knowledge of rare diseases and their treatment.
the paper is interesting and comprehensive, methodologically correct.
Author Response
Response to Reviewer's Comments
Reviewer 1:
we did appreciate your work, in relation to a topic which is still under study, in an era of rapidly evolving knowledge of rare diseases and their treatment.
the paper is interesting and comprehensive, methodologically correct.
We appreciate the comment. Thank you for your kind comment.

Reviewer 2 Report
jpm-1194631
the authors present a review the literature of gynecologic melanomas treated with immunotherapy and discuss the possibility of such treatment in future. One study focusing on the overall survival of gynecologic melanomas separately and five case series and nine case reports concentrating on gynecologic melanomas treated with immune checkpoint inhibitor and/or targeted therapy have been identified and reviewed. The authors conclude that a KIT inhibitor such as imatinib or nilotinib could be the treatment of choice. Moreover, combination of an immune checkpoint inhibitor and a KIT inhibitor may potentially treat cases of resistance to immune checkpoint inhibitors. Major and minor comments can be made.
Major comments
As the authors state in line 360: the diversity of terms referring to gynecologic melanomas is confusing and needs to be more exact. The vulva for example is partly covered by skin and partly covered by mucosa. In the current manuscript it is unclear if melanoma on the cutaneous part of the vulva is included with mucosal melanoma or if this is excluded as cutaneous melanoma. This issue needs to be clarified with care.
The focus of the review is immune checkpoint inhibition therapy, yet the conclusions (lines 365-372) are all on targeted therapy with or without combined immune checkpoint inhibition. The aim and results section should be extensively rewritten to be able to support these conclusions. Otherwise please correct your conclusions to properly reflect the findings of your review.
The discussion is a separate review of KIT inhibition and should be compressed or shifted to the results section with provision of a table that summarizes the findings.
Minor comments
The abstract needs to be rewritten to properly reflect the review as performed in the results section
Please provide differential advise for patients with and without targetable mutations and provide the level of evidence according to the Centre for Evidence-Based Medicine definitions
Author Response
Response to Reviewer's Comments
Reviewer 2:
the authors present a review the literature of gynecologic melanomas treated with immunotherapy and discuss the possibility of such treatment in future. One study focusing on the overall survival of gynecologic melanomas separately and five case series and nine case reports concentrating on gynecologic melanomas treated with immune checkpoint inhibitor and/or targeted therapy have been identified and reviewed. The authors conclude that a KIT inhibitor such as imatinib or nilotinib could be the treatment of choice. Moreover, combination of an immune checkpoint inhibitor and a KIT inhibitor may potentially treat cases of resistance to immune checkpoint inhibitors. Major and minor comments can be made.
Major comments
As the authors state in line 360: the diversity of terms referring to gynecologic melanomas is confusing and needs to be more exact. The vulva for example is partly covered by skin and partly covered by mucosa. In the current manuscript it is unclear if melanoma on the cutaneous part of the vulva is included with mucosal melanoma or if this is excluded as cutaneous melanoma. This issue needs to be clarified with care.
We appreciate the comment. Unfortunately, as almost all papers did not describe definitions of margins between vaginal skin and mucosa referring to vulvar melanoma, we were unable to answer your question. We hope that these definitions will be generalized in the future.
The focus of the review is immune checkpoint inhibition therapy, yet the conclusions (lines 365-372) are all on targeted therapy with or without combined immune checkpoint inhibition. The aim and results section should be extensively rewritten to be able to support these conclusions. Otherwise please correct your conclusions to properly reflect the findings of your review.
We appreciate the comment. We feel sorry for the misleading description; We meant to use the term immunotherapy to refer to both of immune checkpoint inhibitors and targeted therapy. Therefore, this article aims to evaluate both immune checkpoint inhibitors and targeted therapy. To avoid misunderstanding, we have revised the text. We hope this answers your question.
The discussion is a separate review of KIT inhibition and should be compressed or shifted to the results section with provision of a table that summarizes the findings.
We appreciate the comment. We feel sorry for confusing description. Following your advice, we shifted the description of KIT inhibitors to introduction and summarized as a table.
Minor comments
The abstract needs to be rewritten to properly reflect the review as performed in the results section
We appreciate the comment. We rewrote the abstract following as the results section.
Please provide differential advise for patients with and without targetable mutations and provide the level of evidence according to the Centre for Evidence-Based Medicine definitions
We appreciate the comment. In Centre for Evidence-Based Medicine (University of Oxford website), we could not find the answer of your question. In NCCN guideline of cutaneous melanoma (version 2021.2), targeted therapy with BRAF mutations was category 1(high level evidence), but the evidence level of targeted therapy without BRAF mutation was not described. Unfortunately, as for mucosal melanoma (including gynecologic melanoma), we have no answer. We hope this answers your question.

Round 2
Reviewer 2 Report
The manuscript has improved by revision
please explicitly state in your manuscript that for the vulvovaginal melanomas it is not clear whether they represent mucosal or skin melanoma from the studies reviewed. This is of importance since the authors view all gynaecologic melanomas as mucosal melanoma which cannot be deducted from the publications cited.
Minor comment
page 4, line 120: "Netherland" should read: the Netherlands.
Author Response
Response to Editor Comments
Reviewer 2:
please explicitly state in your manuscript that for the vulvovaginal melanomas it is not clear whether they represent mucosal or skin melanoma from the studies reviewed. This is of importance since the authors view all gynaecologic melanomas as mucosal melanoma which cannot be deducted from the publications cited.
Your question about the definition is understandable. It is true that the definition of vulvovaginal melanoma as "a tumor located within X cm of the vulva or vagina is a vulvovaginal melanoma" does not appear in any of the literature on vulvovaginal melanomas. However, we feel sorry for telling you that it would be because that it is a prerequisite that vulvovaginal melanoma is included in the scope of mucosal melanoma. We think it might be a bit short-sighted to assume that one tumor in the vulvovaginal site cannot be distinguished as mucosal or cutaneous melanoma because there is no specific definition of its location. In addition, as mentioned in the Inclusion ad Exclusion Criteria, we excluded gynecologic melanoma present in the vulvovaginal region in metastasis of cutaneous melanoma. Therefore, we recognize all gynecologic melanomas in the literature discussed in this review as mucosal melanoma. However, for the sake of accuracy, we have added the following statement to the site of the Inclusion and Exclusion Criteria. “In the case of vulvovaginal melanomas, one question about the distinction between mucosal and cutaneous melanoma might be occurred. The location of the tumor from the vulva or vagina, specifically within X cm, were not mentioned in any literature. However, as it is assumed that gynecologic melanoma is included in mucosal melanoma in general, we treated all of them as mucosal melanoma.” We hope this answer might fulfill your point.
Minor comment
page 4, line 120: "Netherland" should read: the Netherlands.
We appreciate the comment. We corrected the words.
